# Bi-Objective Dispatch of Multi-Energy Virtual Power Plant: Deep-Learning-Based Prediction and Particle Swarm Optimization

**Jiahui Zhang [1,2]**, **Zhiyu Xu [1,\*]**, **Weisheng Xu [1,3]**, **Feiyu Zhu [1,2]**, **Xiaoyu Lyu [1]** and **Min Fu [1]**

[1]   College of Electronics and Information Engineering, Tongji University, Shanghai 201804, China; karlzhangjh@163.com (J.Z.); xuweisheng@tongji.edu.cn (W.X.); zhufy2015@163.com (F.Z.); 1730751@tongji.edu.cn (X.L.); fumin96704@163.com (M.F.)

[2]   Department of Electrical, Electronic and Information Engineering, University of Bologna, 40136 Bologna, Italy

[3]   Center of Bigdata and Informatization, Tongji University, Shanghai 20092, China

\*   Correspondence: xuzhiyu@tongji.edu.cn; Tel.: +86-137-6449-1271

**Abstract:** This paper addresses the coordinative operation problem of multi-energy virtual power plant (ME-VPP) in the context of energy internet. A bi-objective dispatch model is established to optimize the performance of ME-VPP in terms of economic cost (EC) and power quality (PQ). Various realistic factors are considered, which include environmental governance, transmission ratings, output limits, etc. Long short-term memory (LSTM), a deep learning method, is applied to the promotion of the accuracy of wind prediction. An improved multi-objective particle swarm optimization (MOPSO) is utilized as the solving algorithm. A practical case study is performed on Hongfeng Eco-town in Southwestern China. Simulation results of three scenarios verify the advantages of bi-objective optimization over solely saving EC and enhancing PQ. The Pareto frontier also provides a visible and flexible way for decision-making of ME-VPP operator. Two strategies, "improvisational" and "foresighted", are compared by testing on the Institute of Electrical and Electronic Engineers (IEEE) 118-bus benchmark system. It is revealed that "foresighted" strategy, which incorporates LSTM prediction and bi-objective optimization over a 5-h receding horizon, takes 10 Pareto dominances in 24 h.

**Keywords:** multi-energy virtual power plant; economic cost; power quality; bi-objective dispatch; long short-term memory; multi-objective particle swarm optimization

## 1. Introduction

In recent decades, a worldwide spread of distributed energy resources (DERs) has been evident, such as micro gas engines, wind turbines, photovoltaic panels, small hydropower units, storage devices, electric vehicle charging facilities, etc. However, not only do those emerging technological advances bring us great opportunities of renewable energy exploitation but also enormous challenges in optimal operation. The virtual power plant (VPP) is a promising solution to these issues. The concept of VPP was proposed in late 20th century [1] and applications rapidly spread across Europe and America [2–4]. Based on advanced communication technology and software system, a VPP dispatches a number of DERs, coordinates the operations, and optimizes the overall performance [5]. In China, large-scale VPPs are built along with the dramatic growth of DERs [6]. The VPP standard initiated by the State Grid Corporation of China has been accepted by International Electrotechnical Commission (IEC) in 2018 [7].

A great deal of research concentrates on the optimization of VPP dispatch. Generally, economy is the primary objective, and other factors are also considered in some papers. For example, pollution emissions are taken into account in [8,9] and power quality is set the secondary objective in [10]. On the other hand, various compositions of VPP are investigated. Reference [11] focuses on the performance of VPP integrating electric vehicles and analyzes the impact of access mode. The VPP-based wind-thermal cogeneration is studied in [12]. The coordination of wind power, solar power and pumped storage in a VPP is discussed in [13]. In recent years, the merging technology of combined cooling, heat and power (CCHP) has generated a new member of DER family. The integration of CCHPs results in new challenges due to the interconnection and coupling of multiple energies in forms of electrical, heat and cooling [14,15]. As a sequel, VPPs integrating CCHPs are promoted to multi-energy virtual power plants (ME-VPPs).

The dispatch of ME-VPP is highly challenging mainly due to two difficulties: uncertainties with DERs' and the model's complexity. The prediction of DERs' behavior is vital since it provides input data for the optimization model. Large numbers of efforts are devoted to improve the prediction accuracy and the methods can generally be classified into four categories: (1) Time-series method, which uses sliding mean average of the nearest historical data; (2) Kalman filtering method, which views the historical data as a state variable in a model to predict when the statistic character of noise is known; (3) Statistical method, which realizes parameter estimation and model checking through model identification; (4) Artificial neural networks method, which was made up of multiple neurons to imitate human brain's function and structure [16]. Traditional prediction methods mainly depend on function solving, such as the Grey model [17]. New and more popular prediction methods concentrated on artificial neural network and deep learning [18–20]. In comparison, new methods can fit and learn the historical data about uncertain variables, via which they can achieve more accurate and efficient predictions.

There are plenty of general-purpose commercial optimizers, which are widely used in industry and academic research, such as CPLEX, LINPROG, GUROBI, MOSEK, etc. However, most of them are solely for classical optimization problems with a single objective. Therefore, it is necessary to develop specific algorithms for nontrivial optimization problems with multiple objectives. The chaotic differential evolution is proposed in [21]. An improved genetic algorithm is adopted to optimize VPPs' economic and environmental performances [22]. The multi-objective particle swarm optimization (MOPSO) is an enhanced version of PSO being devoted to multi-objective optimization problems. Considering the efficiency of computation and the simplicity of implementation, MOPSO can be successfully adopted in the field of VPP operation [23,24].

The methods and contributions of this paper are summarized as follows:

(1) A bi-objective optimization model of ME-VPP has been established, which takes economic cost (EC), power quality (PQ), environmental friendliness, and various physical constraints into account.

(2) LSTM, a deep learning method, has been applied to wind prediction. Its superiority to the Grey model has been verified in terms of accuracy, robustness, and computational efficiency.

(3) The realistic case of Hongfeng Eco-town in Southwestern China has been numerically studied by applying MOPSO algorithm. The advantages of bi-objective modeling and "foresighted" strategy have been quantitatively demonstrated.

The remainder of this paper is organized as follows. Section 2 analyzes two objectives of ME-VPP dispatch and establishes the bi-optimization model. Section 3 presents LSTM and MOPSO as the prediction method and the solving algorithm. Computational results of case study and further discussions are demonstrated in Section 4. Finally, Section 5 summarizes the whole work in this paper and draws the conclusions.

## 2. Economy-Quality Bi-Objective Optimization Model

*2.1. Problem Description*

- Model Structure

As depicted in Figure 1, a ME-VPP generally consists of CCHPs, hydro units, wind turbines, photovoltaic panels, storage devices, electric appliances, and cooling/heat facilities. The dispatch of a ME-VPP refers to coordinative operation of these producers and consumers in three forms of energy: electricity, cooling, and heat. In such an interconnected multi-energy carrier framework, CCHPs are particularly significant due to the establishment of coupling among different energy carriers [25].

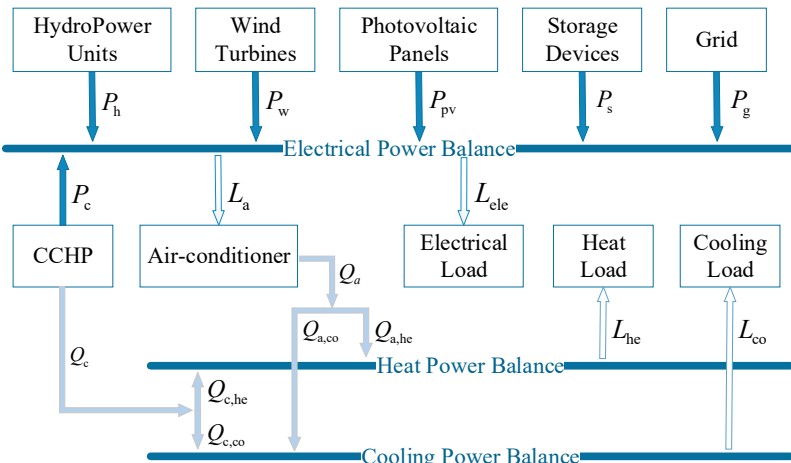

**Figure 1.** Block diagram of multi-energy virtual power plant (ME-VPP).

EC and PQ are two critical indexes evaluating the performance of ME-VPP. An ideal dispatch of ME-VPP is to supply the best PQ by spending the lowest EC. However, better quality generally causes higher cost in practice and vice versa. Considering such an inherent conflicting nature, we employ the framework of multi-objective optimization [9] for problem formulation. A bi-objective dispatch model is established, in which EC and PQ are two independent objectives to be optimized. The Pareto frontier illustrates all noninferior solutions to the dispatch problem.

- Variable Definition

In an ME-VPP, three forms of energy are illustrated by three power buses and each bus is associated with a power balance. As shown in Figure 1, the entity number, electrical energy productions, thermal energy productions, and loads are denoted by $N$, $P$, $Q$, and $L$, respectively. The subscripts specify the entities with which the variables are associated, e.g., 'h' indicates hydropower units and 'w' indicates wind turbine. Variables in Figure 1 can be categorized into two groups: predictable and controllable. The wind power output, solar power output, electrical load, and cooling/heat load are of uncertainty but can be predicted by historical data analysis. The electric/thermal output of CCHP, charge/discharge flow of storage, and exchange volume with main grid are all controllable. In the following subsections, the predictable ones are treated as uncertain external parameters of the bi-optimization model, and the controllable ones are set as the decision variables. The purpose of optimization is to accurately predict external parameters and then to efficiently derive decision variables.

## 2.2. Objectives

2.2.1. Objective 1: To Save Economic Cost

EC of ME-VPP comprises two parts: energy-purchasing cost $\Omega_P$ and environmental governance cost $\Omega_E$.

$$\text{Minimize } F_1 = \Omega_P + \Omega_E, \tag{1}$$

- Energy-purchasing Cost $\Omega_P$

$\Omega_P$ includes cost of fuel consumption of CCHPs and cost of electricity purchased from main grid in case of undersupply.

$$\Omega_P = \sum_{t=1}^{T} \left\{ \sum_{i=1}^{N_c} \left[ f_c(P_c^i(t)) \cdot \lambda_c \right] + P_g(t) \cdot \lambda_g(t) \right\}, \tag{2}$$

Assume the gas (fuel of CCHPs) price $\lambda_c$ is constant, $\lambda_g$ is the electricity price purchasing from the main grid that varies with time and $f_c$ is the quadratic function that calculates the cost of fuel consumption of CCHPs.

$$f_c(P_c^i) = \alpha_2^i \cdot (P_c^i)^2 + \alpha_1^i \cdot P_c^i + \alpha_0^i, \tag{3}$$

where $\alpha_2, \alpha_1, \alpha_0$ are all constant fuel cost factor of each CCHP.

- Environmental Governance Cost $\Omega_E$

$\Omega_E$ reflects the cost of governing pollutant emissions of ME-VPP, which consists of three parts: carbon dioxide ($CO_2$), sulfur dioxide ($SO_2$) and nitrogen oxides ($NO_x$).

$$\Omega_E = \sum_{t=1}^{T} \left\{ \sum_{i=1}^{N_c} \left[ P_c^i(t) \cdot \mu_c \right] + \sum_{j=1}^{N_s} \left[ P_s^j(t) \cdot \mu_s \right] + \sum_{k=1}^{N_g} \left[ P_g^k(t) \cdot \mu_g \right] \right\}, \tag{4}$$

where $\mu_c$, $\mu_s$ and $\mu_g$ denote the pollutant emissions' governance tariffs (unit: CNY/kW·h) of CCHPs, storage devices and the main grid, respectively. And, let $P_g(t) = 0$ if $P_g(t)$ is negative.

2.2.2. Objective 2: To Enhance Power Quality

PQ of ME-VPP involves real power, reactive power, voltage magnitude, voltage angle, etc. This paper mainly concentrates on power index $\Theta_P$ and voltage index $\Theta_U$.

$$\text{Maximize } F_2 = \beta \cdot \Theta_P + (1 - \beta) \cdot \Theta_U, \tag{5}$$

where $F_2$ represents the degree of users' satisfaction. Note that power index of satisfaction degree $\Theta_P$ and voltage index of satisfaction degree $\Theta_U$ are derived from real power losses $P_{loss}$ and voltage stability index $U_\%$ by the fuzzy membership function as follows:

$$\Theta_X = \begin{cases} 0, & X \geq X^0 \\ \dfrac{X^0 - X}{X^0 - X^*}, & X^* < X < X^0, \\ 1, & X \leq X^* \end{cases} \tag{6}$$

where $X \in \left\{ 'P'_{loss}, 'U'_\% \right\}$, $X^0$ and $X^*$ denote initial value and optimal value of $X$ respectively. $\beta$ determines the relative weight between two indexes.

- Real power losses $P_{\text{loss}}$

$$P_{\text{loss}} = \sum_{i=1}^{N_{\text{B}}} \sum_{j=1}^{N_{\text{B}}} G^{ij} \left[ (U^i)^2 + (U^j)^2 - 2U^i U^j \cos \theta^{ij} \right], \tag{7}$$

where $N_{\text{B}}$ represents the number of the buses; $G^{ij}$ denotes the conductance of the transmission line connecting Bus $i$ and Bus $j$; $U^i$ and $U^j$ refer to the voltages of Bus $i$ and Bus $j$; $\theta^{ij}$ denotes the phase angle difference of the voltages of Bus $i$ and Bus $j$.

- Voltage stability index $U_\%$

$$U_\% = \sum_{i=1}^{N_{\text{B}}} \left[ \frac{2U^i - (U^i_{\text{max}} + U^i_{\text{min}})}{2(U^i_{\text{max}} - U^i_{\text{min}})} \right]^2, \tag{8}$$

where $U^i_{\text{min}}$ and $U^i_{\text{max}}$ respectively denote the minimal and maximal voltages of Bus $i$.

*2.3. Constraints*

2.3.1. Constraints 1: Power Balance of ME-VPP

- Electrical Power Balance of ME-VPP

The sum of power output of DERs in ME-VPP should equate with the total electrical load at each time period. Among of all the DERs, $P_{\text{s}}$ and $P_{\text{g}}$ could be negative. Hence,

$$\sum_{i=1}^{N_{\text{h}}} P_{\text{h}}^i(t) + \sum_{j=1}^{N_{\text{w}}} P_{\text{w}}^j(t) + \sum_{k=1}^{N_{pv}} P_{pv}^k(t) + \sum_{m=1}^{N_{\text{c}}} P_{\text{c}}^m(t) + \sum_{n=1}^{N_{\text{s}}} P_{\text{s}}^n(t) + P_{\text{g}}(t) = L_{\text{ele}}(t) + L_{\text{a}}(t), \tag{9}$$

Notice that $L_{\text{a}}$ denotes electrical load of air-conditioners, which is used to satisfy heat/cooling demand.

- Thermal Power Balance of ME-VPP

The sum of heat/cooling power output of DERs in ME-VPP should equate with the total heat/cooling load at each time period. Among of all the DERs, CCHP can generate thermal power simultaneously when it generates electrical power. Rest of thermal power will generate from air-conditioner by transforming from electrical power to thermal power. Hence,

$$\begin{cases} \left( \sum\limits_{i=1}^{N_{\text{c}}} Q_{\text{c},he}^i(t) + Q_{\text{a,he}}(t) \right) = L_{\text{co}}(t) \\ \left( \sum\limits_{i=1}^{N_{\text{c}}} Q_{\text{c},co}^i(t) + Q_{\text{a,co}}(t) \right) = L_{\text{he}}(t) \end{cases}, \tag{10}$$

where $L_{\text{co}}$ and $L_{\text{he}}$ are total cooling load and heat load respectively; $Q_{\text{c}}$ and $Q_{\text{a}}$ denote thermal power generated from CCHP and air-conditioner, which consist of heat power output and cooling power output as $Q_{\text{c,he}}$, $Q_{\text{c,co}}$, $Q_{\text{a,he}}$ and $Q_{\text{a,co}}$.

Thermal power output of CCHP:

$$Q_{\text{c}} = \frac{P_{\text{c}} \cdot (1 - \eta_{\text{e}} - \eta_{\text{L}})}{\eta_{\text{e}}}, \tag{11}$$

where $\eta_e$ is generating efficiency of CCHP, and $\eta_L$ is heat loss coefficient.

$$\begin{cases} Q_{c,he} = Q_c \cdot COP_{c,he} \\ Q_{c,co} = Q_c \cdot COP_{c,co} \end{cases}, \tag{12}$$

where $COP_{c,he}$ and $COP_{c,co}$ are coefficient of CCHPs' performance with respect to heat and cooling energy efficiency.

Thermal power output of air-conditioner:

$$\begin{cases} Q_{a,he} = P_a \cdot COP_{a,he} \\ Q_{a,co} = P_a \cdot COP_{a,co} \end{cases}, \tag{13}$$

where $COP_{a,he}$ and $COP_{a,co}$ are coefficient of air-conditioners' performance with respect to heat and cooling energy efficiency.

### 2.3.2. Constraints 2: Network Transmission Capacity

$$\begin{cases} \Phi^i = U^i \sum\limits_{j=1}^{N_B} U^j \left( G^{ij} \cos \theta^{ij} + B^{ij} \sin \theta^{ij} \right) \\ \Psi^i = U^i \sum\limits_{j=1}^{N_B} U^j \left( G^{ij} \cos \theta^{ij} - B^{ij} \sin \theta^{ij} \right) \end{cases} \quad i = 1, 2, \ldots, N_B, \tag{14}$$

where $\Phi^i$ and $\Psi^i$ refer to real power and reactive power injected into Bus $i$, respectively; $G^{ij}$ and $B^{ij}$ denote conductance and susceptance of the transmission line $(i, j)$ respectively. There are $2N_B$ equations of a power network with $N_B$ buses.

### 2.3.3. Constraints 3: Equipment Output Limits

DERs' upper limits and lower limits:

$$P_{z,min} \le P_z^i(t) \le P_{z,max} z \in \{'h', 'w', 'pv', 'c', 's', 'g'\}, \tag{15}$$

DERs' ramp-rate limits:

$$\left| P_z^i(t) - P_z^i(t-1) \right| \le R_z, \tag{16}$$

where $P_z$ represents the power output of DERs of each kind $z$ and each unit $i$, $R_z$ represents ramp-rate limits of each kind of DERs.

### 2.3.4. Constraints 4: Storage Devices Limits

There are some limitations on charge and discharge rate of storage devices during each time interval, and the amount of charge and discharge is related to the remaining capacity. The following equation and constraints can be expressed for a typical battery [8]:

$$W(t) = W(t-1) + \eta_{ch} P_{ch}(t) \Delta t - \frac{1}{\eta_{dis}} P_{dis}(t) \Delta t, \tag{17}$$

$$W_{min} \le W(t) \le W_{max}, \tag{18}$$

$$0 \le P_{ch}(t) \le P_{ch,max}, \tag{19}$$

$$0 \le P_{dis}(t) \le P_{dis,max}, \tag{20}$$

where $W(t)$ is the amount of energy storage inside the battery at time $t$, $P_{ch}(P_{dis})$ is the permitted rate of charge(discharge) during a definite period of time $\Delta t$, $\eta_{ch}(\eta_{dis})$ is the efficiency of the battery during charge(discharge) process. $W_{min}$ and $W_{max}$ are the lower and upper limits on amount of energy storage

inside the battery and $P_{\text{ch,max}}(P_{\text{dis,max}})$ is the maximum rate of battery charge(discharge) during each time interval $\Delta t$.

For comparison of dispatch result in a single day, we assign the amount of energy storage inside the battery in the end of a day be the same as the beginning.

$$W(0) = W(24), \tag{21}$$

## 3. Method of Prediction and Optimization

### *3.1. Long Short-Term Memory (LSTM)*

Uncertain variables such as power output and load play a significant role during the operation of ME-VPP, and they can affect the accuracy and efficiency of power management directly. Thus, powerful prediction method and appropriate prediction time horizon are necessary for the operation of ME-VPP. A plenty of prediction methods were proposed to predict the uncertain variables, such as traditional method (Grey model, Kalman, etc.) and novel method (RNN, LSTM, etc.). The prediction method should meet the accuracy and efficiency of operation work, meanwhile, a proper prediction time horizon should be selected. If too short, system operator may fail in sufficient preparation for power management in few hours later; if too long, computation resources will be wasted and the accuracy of predicted variables cannot be guaranteed.

In this paper, we choose the LSTM to predict the uncertain variables. Corresponding experiments are set in Section 4 to choose the proper time horizon of prediction.

#### 3.1.1. LSTM Introduction

In this paper, we employ LSTM to understand complex fluctuation of curves and generate predicted curves about incoming several hours. Due to the shortcoming of RNN, such as vanishing and exploding gradient problem, LSTM was developed to analyze and predict more accurately and efficiently [26]. We briefly introduce the structure of LSTM in this section. LSTM has a memory called "cell", which is used to store the state vector summarizing the sequence of the past input data. The state of the cell is updated by input data, output data and previous state of cell. Assume $c_t$ denote the state of the memory cell at time $t$. Then, $c_t$ in a LSTM model can be updated by the following equations recursively.

$$i_t = \sigma\left(W_{x,i}x_t + W_{h,i}h_{t-1} + b_i\right), \tag{22}$$

$$f_t = \sigma\left(W_{x,f}x_t + W_{h,f}h_{t-1} + b_f\right), \tag{23}$$

$$o_t = \sigma\left(W_{x,o}x_t + W_{h,o}h_{t-1} + b_o\right), \tag{24}$$

$$g_t = \tanh\left(W_{x,c}x_t + W_{h,c}h_{t-1} + b_c\right), \tag{25}$$

$$c_t = f_t \otimes c_{t-1} + i_t \otimes g_t, \tag{26}$$

$$h_t = o_t \otimes \tanh(c_t), \tag{27}$$

where $\sigma(x) = \frac{1}{1+e^{-x}}$ is sigmoid function; $x \otimes y$ denote element-wise product; $W_{x,i}$, $W_{h,i}$, $W_{x,f}$, $W_{h,f}$, $W_{x,o}$, $W_{h,o}$, $W_{x,c}$, $W_{h,c}$ are weight matrix for linear transformation; $b_i$, $b_f$, $b_o$, $b_c$ are bias vectors; three gating vectors $i_t$, $f_t$, $o_t$ denote input, forget, and output respectively; $g_t$ is the state update vector and $h_t$ is the output hidden state vector.

Figure 2 depicts the basic structure of LSTM. The zero configuration of $f_t$ can let the network forget the information $c_{t-1}$ stored in the memory cell. The input gate $i_t$ and the output gate $o_t$ can control the information flow from the input to the output. Note that the state of gate is learned from training data. In order to acquire the information corresponding to the given target, we add an additional output network to the hidden state $h_t$ [27].

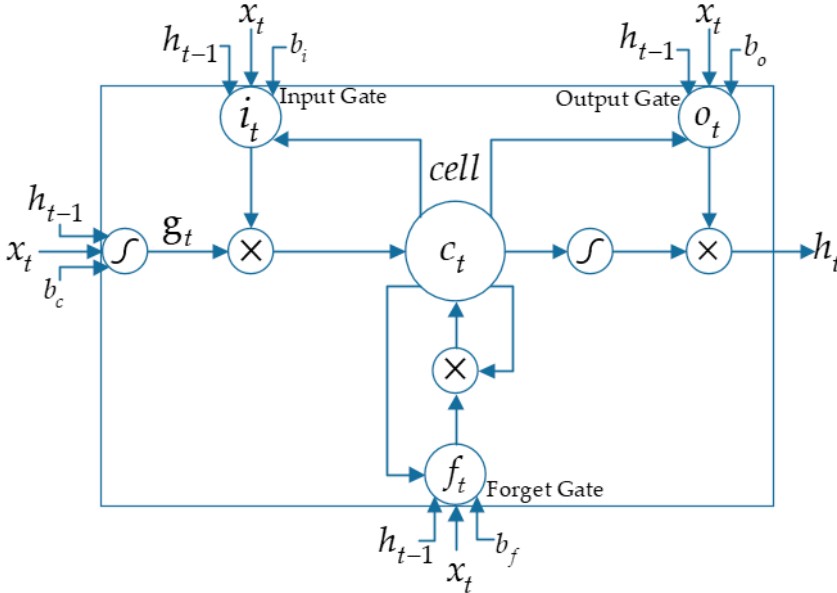

**Figure 2.** Basic structure of long short-term memory (LSTM).

### 3.1.2. Model Training

Curves of uncertain variables which should be learnt and predicted include wind power, solar power and three types of load (electrical load, heat load and cooling load). Each of them needs a specific LSTM prediction model, we take the wind prediction as an example in this paper. Other uncertain variables could be predicted by the same method.

Without loss of generality, we take the LSTM model for wind prediction for instance. As in Figure 3, the total wind power at the time step $t$ is denoted as $x(t)$. The sequence of the coordinates $x(t-n), \cdots, x(t)$ taken from $n+1$ hours can be labeled as the training data. Then, $x(t-n+1), \cdots, x(t+1)$ can be labeled as next sequence. A plenty of sequences involve several years form into training set to fit the LSTM model, where the last data in every sequence are the learning targets of the training data. In this paper, we orderly set input fully-connected layers, two LSTM cells followed by dropout layers respectively, and the output fully-connected layers. "Adam" optimizer and "mean absolute error" loss standards are used to compile the model.

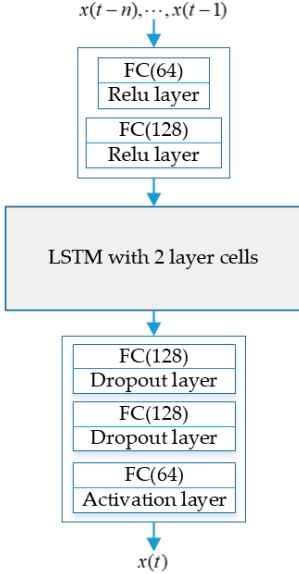

**Figure 3.** Structure of LSTM for wind prediction.

### 3.1.3. Prediction Procedure

Prediction procedure can be simply concluded as two steps: Input data preparation and inference by prediction model.

- Input data preparation

Firstly, we should extract valid information from historical data. Then extracted data need to be formatted as specific tensor. Taking account of the different time horizons of prediction, every step moving forward should replace the last input data by the predicted data. In this paper, we set the 24 experiments of different prediction time horizon ahead from 15 min to 6 h, their time intervals are 15 min.

- Inference by prediction model

The prediction model fit by aforementioned method can be regarded as a function constituted by neural network. Input prepared data to the model can get the predicted data. Experiments result about different time horizons of prediction will be presented in the next section.

### 3.2. Multi-Objective Particle Swarm Optimization (MOPSO)

### 3.2.1. MOPSO Introduction

Pareto optimality is a classical method to optimize conflicting objectives which is impossible to make any individual or preference criterion better off without making at least one individual or preference criterion worse off. The result of Pareto optimality is a set of Pareto front, which is known as the Pareto frontier. The algorithm we applied in this paper is MOPSO, which is used to find the Pareto frontier.

MOPSO is a popular algorithm in the field of real parameters processing and conflicting objectives optimization. It is derived from the classical PSO [28], which is based on the update formula:

$$V^{y+1} = \omega \times V^y + C_1 \times rand_1(\cdot) \times (pbest - X^y) + C_2 \times rand_2(\cdot) \times (gbest - X^y) \\ X^{y+1} = X^y + V^{y+1} \qquad , \qquad (28)$$

where $X$, $V$ denote the position and velocity of a particle in the solution space, respectively; *pbest*, *gbest* denote the personal best and the global best of $X$, respectively; $\omega$, $C_1$ and $C_2$ denote the velocity update rate, the *pbest*-based position evolution rate, and the *gbest*-based position evolution rate, respectively. *y* in superscript is the generation index. MOPSO evolves from (28), but it has two fitness functions and a set of Pareto optimality standards. In each generation, MOPSO not only updates particles' position and velocity, but also the Pareto frontier.

### 3.2.2. Improved Parts and Scheme

MOPSO designed in this paper could tackle the problem mentioned above precisely and efficiently. In order to better deal with the preceding optimization problem, this paper mainly modified frontier update and *gbest* selection. For frontier update, MOPSO adopts double refinement of frontier in each generation. Non-inferior solutions can enter Pareto frontier in the first refinement, then frontier weeds out some inferior solutions with respect to other solutions in the second refinement. For *gbest* selection, MOPSO adopts changing tactic that allocates particles to each *gbest*, which have two periods of time. It starts with allocating particles evenly to each *gbest* in order to keep the diversity of swarm. It would select *gbest* randomly from frontier for each particle after several generations, thus, particles will swarm to the concentrated area by reason of possibility. In all, this method pursues diversity in the early period and seeks high speed of convergence in later period [9].

MOPSO can be split into three parts: main function, fitness function, constraint function. Main function manages the logic of each evolution of swarm. Fitness function calculates objectives

value such as cost of operation and the degree of users' satisfaction about power quality. Constraint function ensures the three equality constraints and two inequality constraints. Detailed scheme outlines as Figure 4.

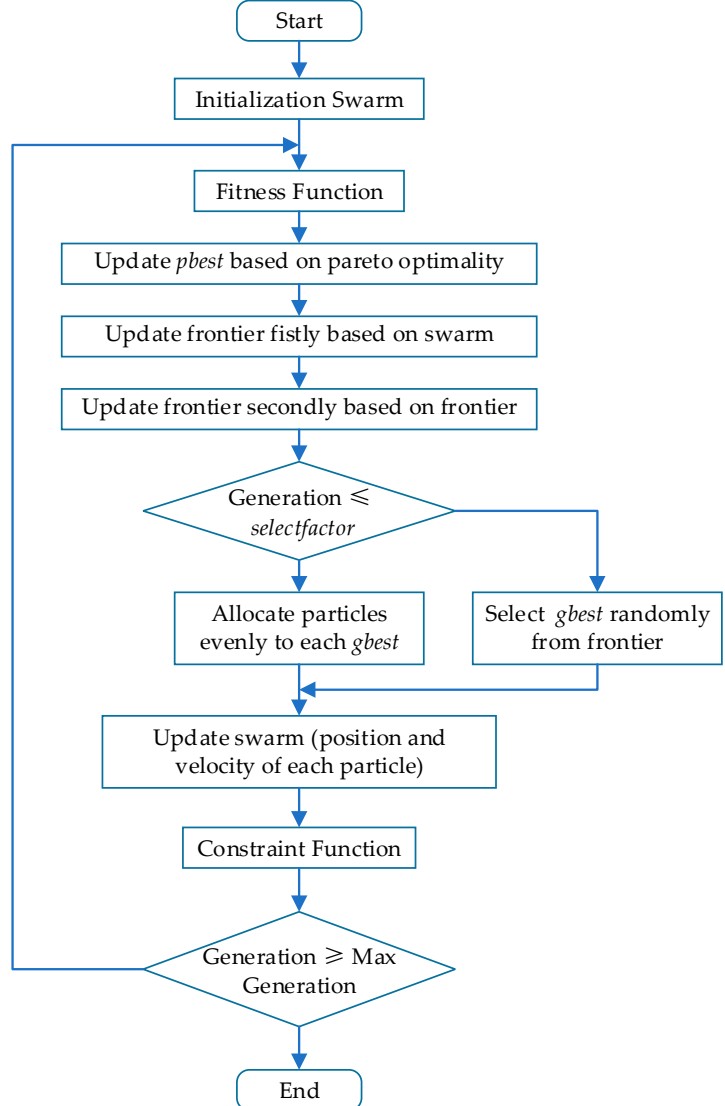

**Figure 4.** Scheme of improved multi-objective particle swarm optimization (MOPSO).

## 4. Case Study

### 4.1. Case Description

4.1.1. ME-VPP of Hongfeng Eco-Town in Southwestern China

Hongfeng Eco-town is a demonstration base operated by Guizhou Power Grid Corporation for multi-energy coordinative dispatch. As shown in Figure 5, it is located on the shore of Hongfeng Lake, which is 27 km from Guiyang, a capital city of Southwestern China. Covering a terrain of 3.35 km$^2$, Hongfeng Eco-town is equipped with a hydropower unit, a CCHP, three wind turbines, three photovoltaic (PV) arrays, two battery blocks (lithium iron phosphate), and an electric vehicle (EV) charging station. The network structure is shown in Figure 6 and parameters are listed in Table 1 [9].

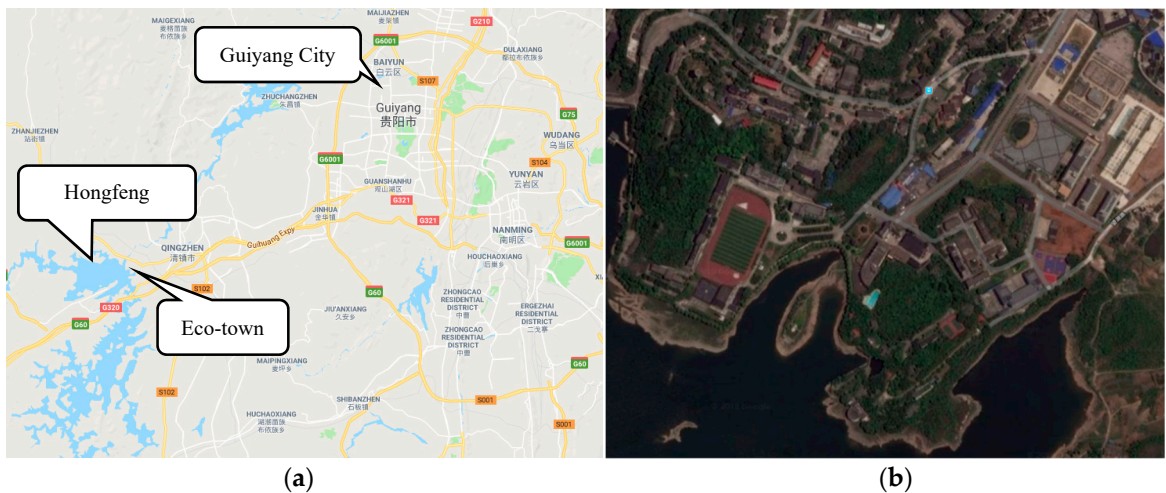

**Figure 5.** Hongfeng Eco-town on Google Maps: (**a**) geographical location; (**b**) satellite image.

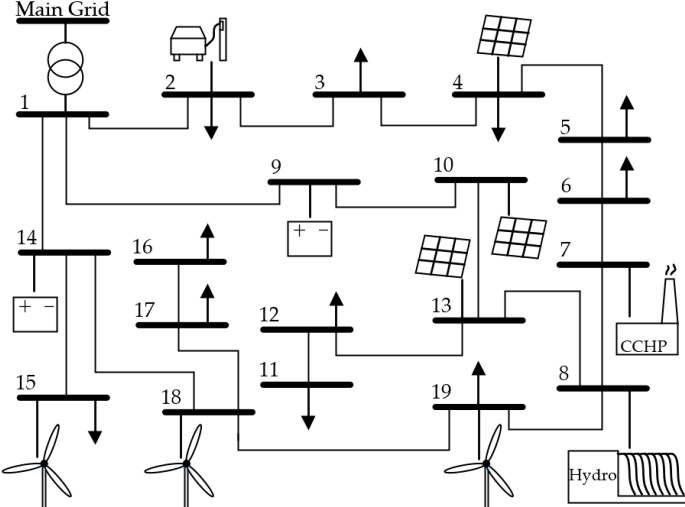

**Figure 6.** One-line diagram of ME-VPP in Hongfeng Eco-town.

**Table 1.** Bus parameters of ME-VPP in Hongfeng Eco-town.

| Bus | Facilities | Power Source | | Power Load |
|:---:|:---:|:---:|:---:|:---:|
| | | Lower Limit (MW) | Upper Limit (MW) | Nominal (MW) |
| 1 | Main grid | −10 | 10 | - |
| 2 | EV charging station & load | −0.43 | 0.43 | 1.12 |
| 3 | load | - | - | 0.52 |
| 4 | PV array & load | 0 | 0.61 | 1.41 |
| 5 | load | - | - | 0.37 |
| 6 | load | - | - | 4.35 |
| 7 | CCHP | 0 | 2.86 | - |
| 8 | Hydropower unit | 0 | 10 | - |
| 9 | Battery block & load | −0.43 | 0.43 | 1.02 |
| 10 | PV array& load | 0 | 0.61 | 1.96 |
| 11 | Load | - | - | 1.91 |
| 12 | Load | - | - | 2.65 |
| 13 | PV array | 0 | 0.61 | - |
| 14 | Battery block | −0.43 | 0.43 | - |
| 15 | Wind turbine & load | 0 | 0.69 | 4.22 |
| 16 | Load | - | - | 1.55 |
| 17 | Load | - | - | 3.97 |
| 18 | Wind turbine | 0 | 0.69 | - |
| 19 | Wind turbine & load | 0 | 0.69 | 0.25 |

Without loss of generality, we randomly select certain day in Eco-town as a typical day for research. The electrical load, power output of wind turbines and photovoltaic panels are drawn as Figure 7. As a multi-energy carrier system, CCHP is researched by annual operating model generally [25,29], since heat load mainly exists in winter and cooling load mainly exists in summer. Few cooling loads in winter and heat loads in summer could be supplied by electrical devices that require electrical load. Three types of load in a typical day are depicted in Figure 8.

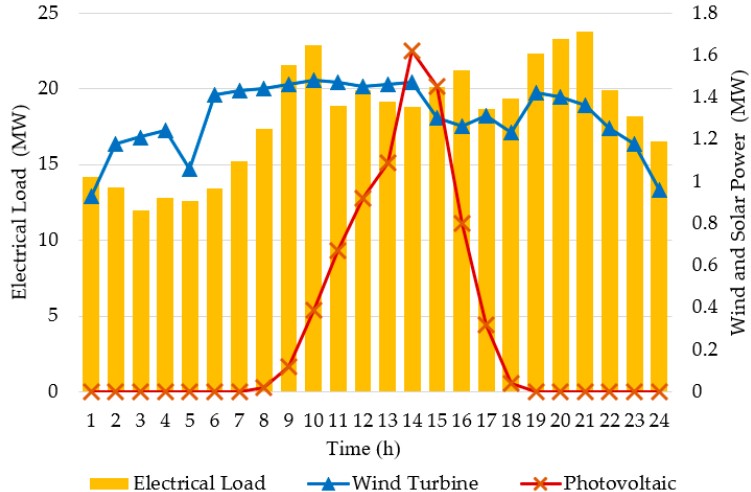

**Figure 7.** Profiles of wind power, solar power and electrical load in a typical day.

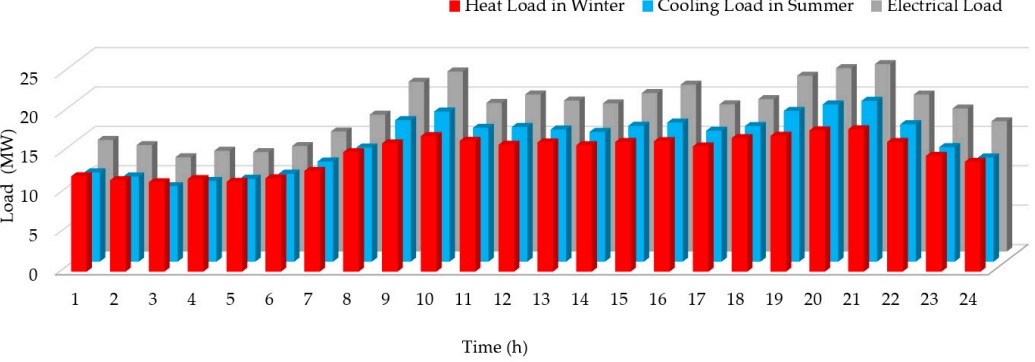

**Figure 8.** Three types of load in a typical day.

In order to calculate the cost of environmental governance, some pollutant emissions coefficient and cost about China are searched and listed as Table 2. Appropriate adjustments have been made according to the energy particularity of Guizhou province. The row of $\mu$ records environmental governance tariffs of CCHP, storage devices, and the main grid, which are $\mu_c$, $\mu_s$, and $\mu_g$, respectively.

**Table 2.** Pollutant emissions coefficient and environmental governance tariff.

| Pollutant Emissions | Coefficient of Emission kg/MW·h | | | Governance Tariff (CNY/kg) |
|---|---|---|---|---|
| | CCHP | Storage Devices | Main Grid | |
| $CO_2$ | 408.6 | 25.8 | 778.5 | 0.210 |
| $SO_2$ | 0.056 | 0.0003 | 0.397 | 14.843 |
| $NO_x$ | 0.1500 | 0.0075 | 0.3410 | 62.964 |
| - | Governance Tariff (CNY/kW·h) | | | - |
| $\mu$ | 96.08 | 5.89 | 190.85 | - |

### 4.1.2. Simulation Environment

- Hardware: CPU: 8 Intel (R) Core (TM) i7-7700 @ 3.6GHz; GPU: GeForce GTX 1050.
- Operating system: Windows 10 64-bit; Ubuntu 16.04.
- Software tools: MATLAB-2017a; Keras-2.2.2; Tensorflow-1.10.0; Python-2.7.12; Cuda-9.0; Cudnn-7_cuda9.0.

### 4.2. *LSTM-Based Wind Prediction*

#### 4.2.1. Training Data and Prediction Results

Historical data of wind power of nine years (2009–2017) are utilized as the training set of the LSTM model. Twenty-four hours of data of wind power randomly selected in 2018 is utilized as the validation set. Prediction step is one of the most significant parameters of LSTM prediction model. Aiming to improve the prediction accuracy, 24 different prediction steps were tested: from 15-min ahead to 6-h ahead. We take 1 h as interval to show the prediction result, and take 15 min as interval to further comparison in the following subsection. In Figure 9, the real data of wind power are depicted with the thick dashed line, and the predicted data of wind power are depicted with thin solid lines. Notice that the colors are gradually getting darker as the shortening of prediction step. Intuitively, the shortest prediction step can achieve the highest prediction accuracy.

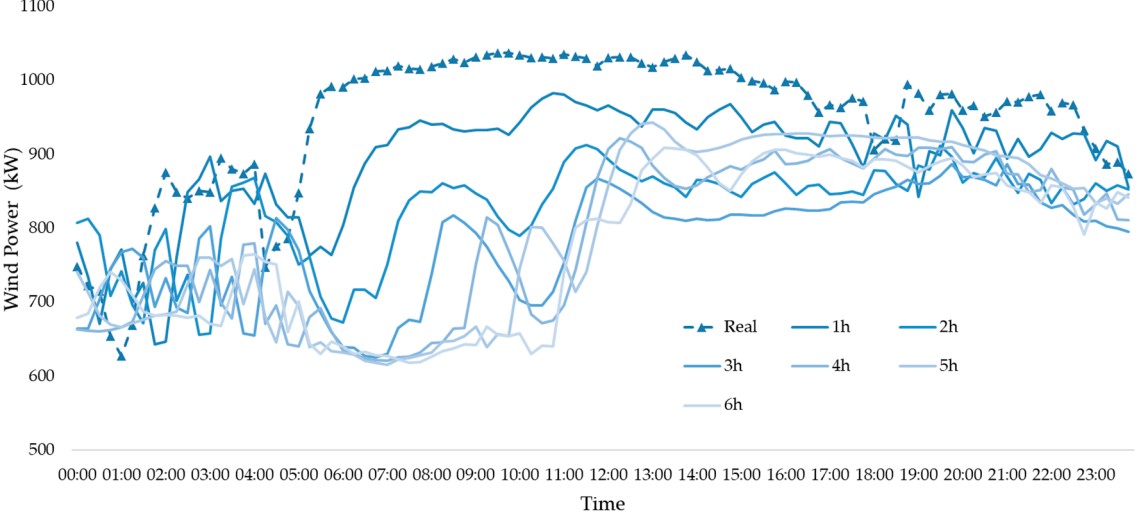

**Figure 9.** Prediction results of LSTM.

#### 4.2.2. Comparison with Grey Model Prediction

The same experiment is conducted on Grey model, and the prediction results with 24 different prediction steps are depicted in Figure 10. We compare the performance of LSTM with Grey model by root mean square error (RMSE) and maximum error (ME). As depicted in Figure 11, LSTM prediction has smaller RMSEs in most cases of 24 prediction steps, and MEs of LSTM prediction are always smaller than those of Grey model. Besides, the trajectories of LSTM prediction are more stationary.

Furthermore, the computing time of the LSTM model is approximately 50 times shorter than that of the Grey model (hundreds of milliseconds versus tens of seconds). Since the sophisticated process of deep learning on historical data is conducted offline and the neural network is pre-trained, the prediction of LSTM is as simple as invoking a function. On the contrary, every prediction of Grey model involves solving a number of complicated equations.

To summarize, LSTM prediction has advantages over the Grey model in terms of accuracy, robustness and computational efficiency.

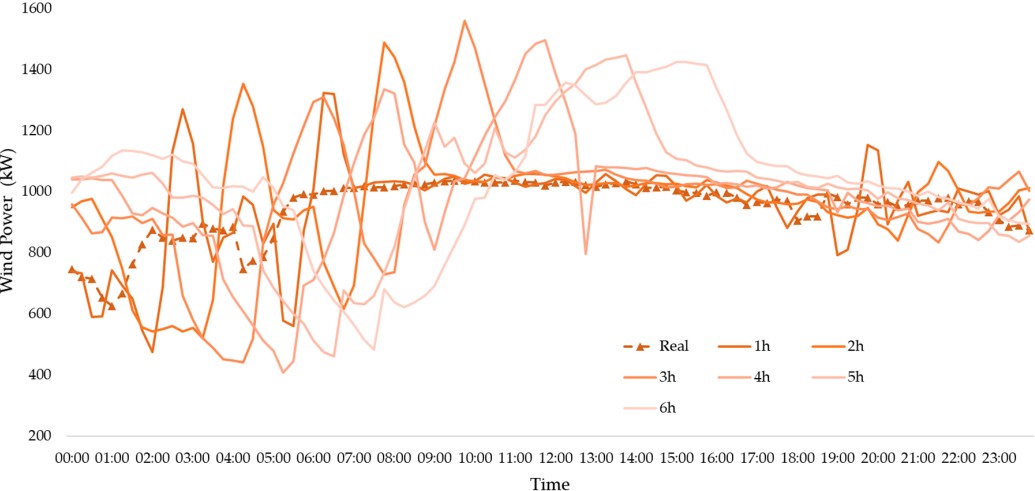

**Figure 10.** Prediction results of Grey model.

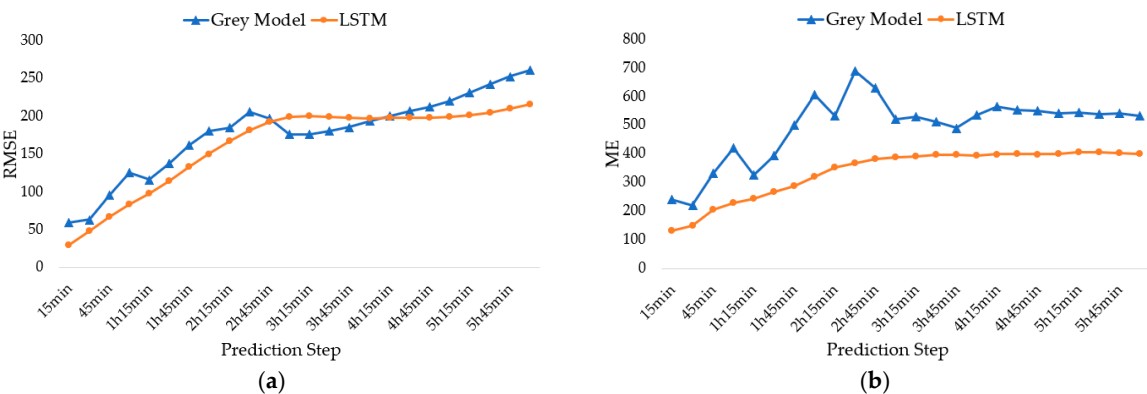

**Figure 11.** Comparison of prediction errors of Grey Model and LSTM: (**a**) root mean square error; (**b**) maximum error.

### 4.3. MOPSO-Based Problem Solution

#### 4.3.1. Comparison of Bi- and Mono-Objective Modeling

In order to verify the effectiveness of bi-objective modeling and the advantages over the mono-objective counterparts, we investigate the following three scenarios:

- Scenario A: "Money-oriented", a mono-objective model that solely saves EC.
- Scenario B: "User-first", a mono-objective model that solely enhances PQ.
- Scenario C: "Pareto-optimal", a bi-objective model that compromises EC and PQ.

The running data of Hongfeng Eco-town in a typical day are utilized for simulation. The scenarios formulated by mono-objective models are solved by using classical PSO algorithm, and Scenario C with a bi-objective model is solved by using MOPSO algorithm. The particle population is 200 and the maximal generation is 150. Each scenario is tested by 20 trials of simulation, and the optimization processes are depicted in Figure 12.

In Scenario A, EC monotonically decreases with the generation counts since the optimization is "money-oriented". On the other hand, the curves of PQ are fluctuant and eventually converge to low levels. It is revealed that pursuing economies blindly will face the risk of losing users' satisfaction upon power quality. On the contrary, the PQ curves of Scenario B are monotonically increasing due to the "user-first" doctrine in optimization. Nonetheless, higher cost is inevitable. As a result, it is necessary to consider Scenario C, which reconciles the two conflicting objectives and derives the Pareto-optimal solutions.

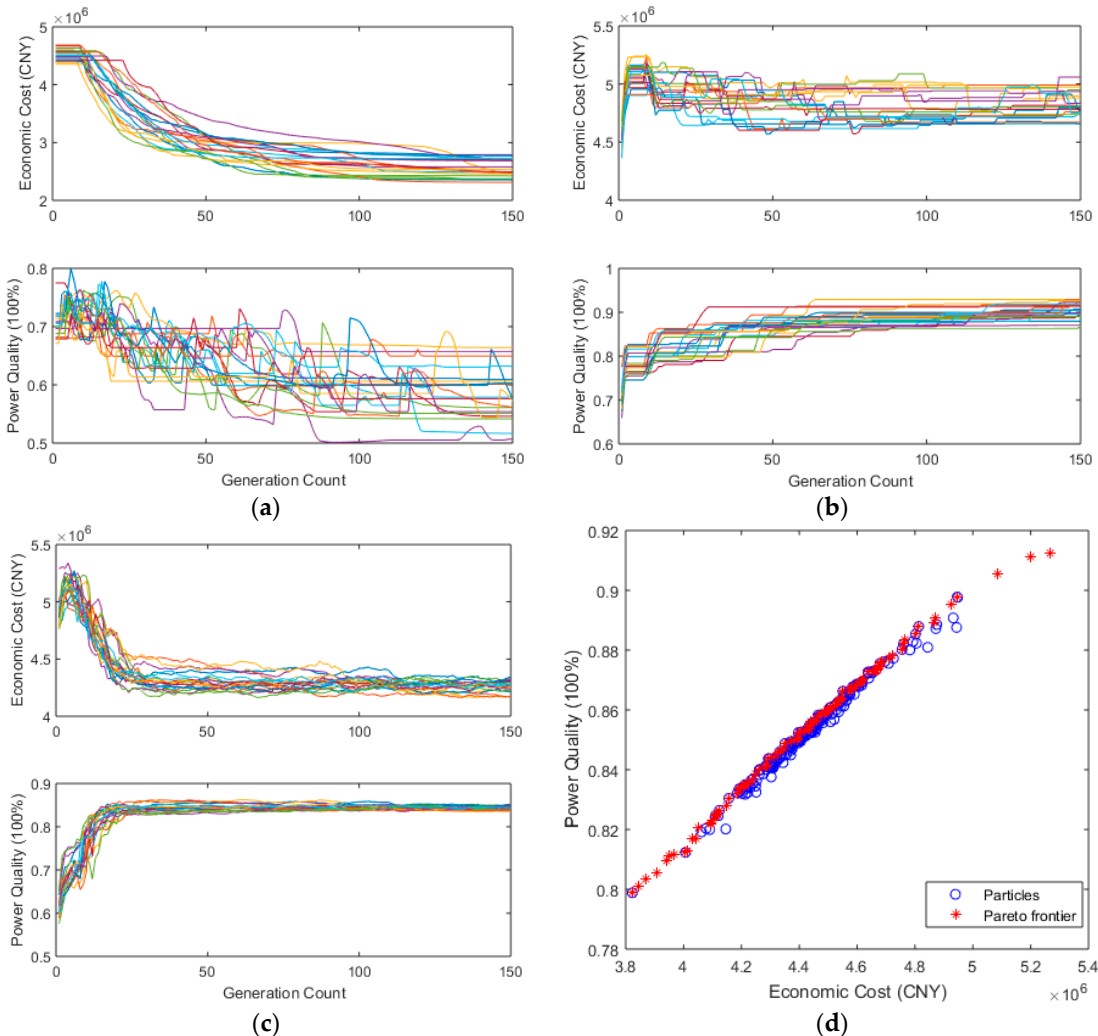

**Figure 12.** Optimization results of mono-objective and bi-objective: (**a**) "money-oriented" scenario; (**b**) "user-first" scenario; (**c**) "Pareto-optimal" scenario; (**d**) economic cost and power quality (EC-PQ) plane after 150 generations.

Figure 12d demonstrates the converged positions of 200 particles in EC-PQ plane after 150 generations of optimization. Each particle of Pareto frontier, which is denoted by a red star, is a non-inferior solution to the bi-objective optimization problem. Without loss of generality, one Pareto optimum is selected from the median position of Pareto frontier and its convergence history is depicted in Figure 12c. Neither EC nor PQ are monotonic functions of generation count due to the metaheuristic nature of MOPSO. However, in a global view, EC and PQ respectively converge to low and high levels, i.e., EC and PQ are simultaneously improved. Numeric results of three scenarios are listed in Table 3. The Pareto frontier provides a visible and flexible way for decision guidance. The operator ME-VPP can select dispatch solution from Pareto frontier according to various specifications of scenarios. One noninferior solution of the Pareto frontier is shown in Figure 13, which contains power output and the amount of energy storage inside the storage devices.

**Table 3.** Optimization results of three scenarios.

| Scenario | Description | Optimization Objective | EC ($10^6$ CNY) | PQ (Score in 100) |
|----------|-------------|------------------------|-----------------|-------------------|
| A | Money-oriented | Economic cost (EC) | 2.55 | 58.30 |
| B | User-first | Power quality (PQ) | 4.82 | 90.38 |
| C | Pareto-optimal | EC and PQ | 4.26 | 84.36 |

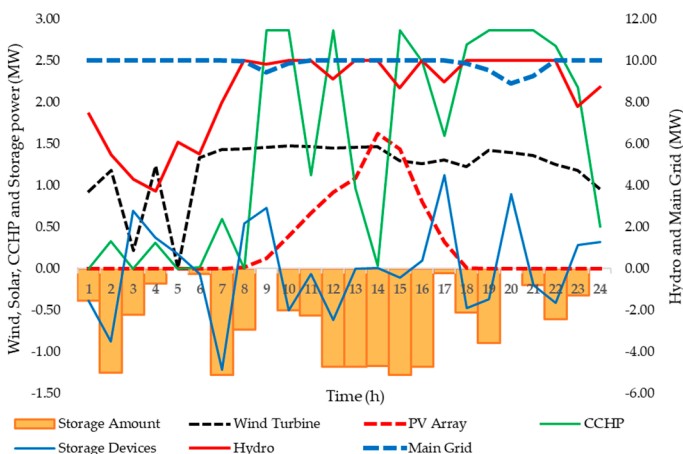

**Figure 13.** One noninferior solution of the Pareto frontier—power production/consumption (MW) in ME-VPP in 24 h.

### 4.3.2. Discussion on Receding Horizon of Optimization

This subsection tests the overall performance of the proposed bi-objective dispatch of ME-VPP. The mechanism of receding horizon is employed to incorporate the predicted external parameters (e.g., wind power) and the optimized decision variables (e.g., CCHP output). We investigate two strategies as follows:

- Strategy A: "Improvisational", which operates ME-VPP on a 1-h receding horizon.
- Strategy B: "Foresighted", which operates ME-VPP on a 5-h receding horizon.

The flow diagrams of two strategies are illustrated in Figure 14. It should be emphasized that two strategies both operate ME-VPP in a 1-h-ahead framework, i.e., the bi-objective optimization problem is solved every hour and the dispatch solution is applied next hour. But they are discriminated by different prediction steps and optimization horizon: the "improvisational" strategy incorporates 1-h wind prediction result and only derives the dispatch solution of the next one hour; "foresighted" strategy utilizes 5-h wind prediction and computes dispatch solution for the next five hours (although only the first one hour is actually applied).

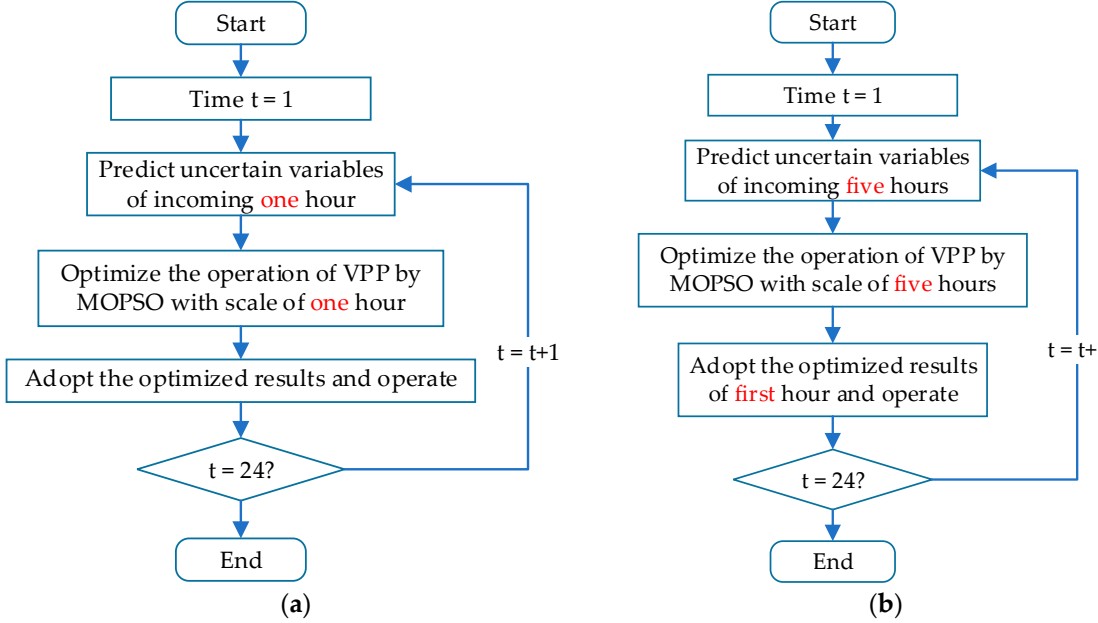

**Figure 14.** Schemes of two strategies: (**a**) improvisational; (**b**) foresighted.

Due to the relatively small scale of Hongfeng Eco-town, the IEEE 118-bus system is adopted as the benchmark to test the overall performances of the two aforementioned strategies. The IEEE 118-bus system is a large-scale and complex case consisting of 118 buses, 186 branches, 91 loads and 54 generators [30]. The simulation is performed by using 24 h of data of a typical day. The particle population is 200 and the maximal generation number is 50.

The overall performance is quantified by EC and PQ. Each solution corresponds to a point on the EC-PQ plane and all optimal solutions forms the Pareto frontier. In each hour, the numeric simulation obtains two Pareto frontiers on EC-PQ plane. Each Pareto frontier corresponds to one strategy. For the ease of comparison, we select the two closest points on the EC-PQ plane, each from a Pareto frontier. These two points represents the performances of two strategies in each hour, and 24-h comparison results are listed in Table 4.

**Table 4.** Comparison of Pareto-optima of two strategies: "improvisational" and "foresighted".

| Time | "Improvisational" | | "Foresighted" | | Improvement of "Foresighted" | | |
|------|-------------------|--|---------------|--|------------------------------|--|--|
| | EC ($10^3$ CNY) | PQ (Score in 100) | EC ($10^3$ CNY) | PQ (Score in 100) | Saved EC ($10^3$ CNY) | Enhanced PQ (Score in 100) | Pareto Dominance |
| 1 | 268.63 | 78.30 | 257.53 | 82.38 | 11.10 | 4.08 | Yes |
| 2 | 223.31 | 60.53 | 216.35 | 78.16 | 6.96 | 17.63 | Yes |
| 3 | 225.25 | 89.96 | 237.96 | 91.23 | −12.71 | 1.27 | − |
| 4 | 247.93 | 78.20 | 245.68 | 93.65 | 2.24 | 15.46 | Yes |
| 5 | 205.90 | 68.73 | 203.23 | 86.87 | 2.67 | 18.14 | Yes |
| 6 | 219.07 | 66.29 | 218.30 | 77.37 | 0.78 | 11.08 | Yes |
| 7 | 181.42 | 72.49 | 176.20 | 89.20 | 5.22 | 16.71 | Yes |
| 8 | 240.80 | 73.49 | 217.44 | 84.80 | 23.35 | 11.31 | Yes |
| 9 | 318.97 | 62.33 | 348.50 | 61.50 | −29.52 | −0.83 | No |
| 10 | 263.16 | 73.22 | 281.84 | 77.10 | −18.68 | 3.88 | − |
| 11 | 210.04 | 73.83 | 190.22 | 92.90 | 19.82 | 19.07 | Yes |
| 12 | 261.44 | 72.11 | 265.77 | 92.86 | −4.33 | 20.76 | − |
| 13 | 240.77 | 65.14 | 226.19 | 87.93 | 14.58 | 22.79 | Yes |
| 14 | 240.53 | 86.33 | 249.50 | 88.21 | −8.97 | 1.88 | − |
| 15 | 233.62 | 67.91 | 223.65 | 68.76 | 9.98 | 0.86 | Yes |
| 16 | 327.05 | 71.12 | 348.67 | 77.43 | −21.62 | 6.31 | − |
| 17 | 351.05 | 69.08 | 367.12 | 87.96 | −16.07 | 18.88 | − |
| 18 | 254.36 | 58.24 | 275.00 | 60.44 | −20.64 | 2.19 | − |
| 19 | 312.58 | 57.78 | 342.19 | 61.87 | −29.62 | 4.08 | − |
| 20 | 307.64 | 68.44 | 334.02 | 77.42 | −26.37 | 8.97 | − |
| 21 | 246.52 | 70.49 | 286.10 | 57.02 | −39.58 | −13.47 | No |
| 22 | 265.93 | 77.17 | 277.21 | 86.55 | −11.29 | 9.37 | − |
| 23 | 264.82 | 71.48 | 273.59 | 59.35 | −8.77 | −12.13 | No |
| 24 | 269.26 | 69.17 | 223.93 | 72.71 | 45.34 | 3.54 | − |
| **Mean** | - | **70.91** | - | **78.90** | - | **7.99** | - |
| **Sum** | **6180.05** | - | **6286.19** | - | **−106.14** | | - |

There are four major columns in Table 4. The first indicates the sequence of 24 h in a day. The second and third are the selected Pareto optima of "improvisational" and "foresighted", respectively; each Pareto optimum represents the strategy performance in terms of EC and PQ. The fourth major column demonstrates the improvement of "foresighted" strategy. Two indexes, "saved EC" and "enhanced PQ", are investigated with respect to the "improvisational" strategy. The "foresighted" strategy has Pareto dominance if both indexes are positive numbers, i.e., achieving higher quality with lower cost. Otherwise "improvisational" strategy takes Pareto dominance if both indexes are negative numbers; and neither one takes Pareto dominance if one index is positive and the other is negative. Table 4 demonstrates that the "improvisational" and "foresighted" strategies take Pareto dominance in 3 h and 10 h of one day, respectively.

The last two rows at the bottom of Table 4 further compare the two strategies in terms of the total EC and average PQ in 24 h. It is shown that neither takes Pareto dominance. Nevertheless, the "foresighted" strategy can achieve 11.27% higher quality than the "improvisational" strategy by spending 1.69% more cost ($106.14 \times 10^3$ CNY in 24-h).

## 5. Conclusions

This paper investigates the multi-energy interconnection, interaction and coordination in ME-VPP. A bi-objective optimization problem is formulated, which simultaneously saves economic cost (EC) and enhances power quality (PQ). EC consists of energy-purchasing cost and environmental governance cost, and PQ is the weighed aggregation of voltage stability and active power loss. Various realistic factors are considered, which include multi-energy coupling, pollutant emission tariff, power balance of buses, transmission capacity, equipment output limits, etc.

A realistic case of Hongfeng Eco-town in Southwestern China is studied. LSTM is adopted for wind prediction. Based on the deep-learning of nine-year historical data, we tested LSTM performances on various prediction steps: from 15-min ahead to 6-h ahead. Numeric results demonstrate that LSTM is superior to the traditional grey model in terms of prediction accuracy, robustness, and computational efficiency. MOPSO is applied as the solving algorithm, and three scenarios are investigated: "money-oriented", "user-first", and "Pareto-optimal". It is verified that the bi-objective modeling is capable of compromising two objectives, as compared with the mono-objective counterparts.

Two strategies—"improvisational" and "foresighted"—were proposed to test the overall performances. Both incorporate LSTM wind prediction and bi-objective optimization, but are over different receding horizons: 1-h and 5-h, respectively. Test results on IEEE 118-bus system indicated that the "foresighted" strategy takes much more time of Pareto dominance in a day.

**Author Contributions:** Conceptualization, Z.X. and W.X.; Data curation, F.Z.; Funding acquisition, Z.X. and W.X.; Methodology, J.Z.; Resources, M.F.; Software, J.Z. and F.Z.; Supervision, W.X.; Validation, X.L.; Writing—original draft, J.Z.; Writing—review & editing, Z.X.

**Funding:** This research was funded by National Natural Science Foundation of China, grant number 61773292, 71401125; Research & Development Program of Guizhou Power Grid Co., Ltd., grant number GZKJXM20160635.

**Acknowledgments:** The authors gratefully acknowledge Jing Nong of Guizhou Power Grid Co., Ltd. for the technical seminars. Special thanks to system operators for providing historical data of Hongfeng Eco-town. The authors greatly appreciate the two autonomous reviewers for their constructive suggestions that make the manuscript improved. Sincere gratitude also goes to Jinzhi Kuang and Zhoujie Ma of the University of Sydney for their elaborate and timely helps in proofreading.

**Conflicts of Interest:** The authors declare no conflicts of interest.

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
