# Peer review of "Bi-Objective Dispatch of Multi-Energy Virtual Power Plant: Deep-Learning-Based Prediction and Particle Swarm Optimization"

_applsci, doi:10.3390/app9020292_

Reviewer 1 Report

The paper presents a method for the solution of the Bi-Objective Dispatch of Multi-Energy Virtual Power Plants (VPP) based on deep learning based forecasting (Long short-term memory (LSTM)) and  improved multi-objective particle swarm optimization (MOPSO).  The paper addresses a timely and interesting topic.   The presentation is clear, although it suffers from scattered language errors, and the test results with real data from an existing VPP in China are interesting.

The model presented is sufficient with the exception of the modeling of storage.   The storage capacity (MWh) is not modeled at all (nor relevant data are presented in the results), the storage dynamics are ignored from the model, thus making the results regarding storage highly questionable.   What is the storage policy used?  How is it decomposed in 1h and 5h segments in the "improvisational" and "foresighted" strategies?  These are questions that the authors must address for their storage mode to be meaningful.  In addition, the pollution footprint of storage is not correctly calculated.  A correct calculation of storage related emissions is to compute the increase of emissions during charging and deduct the decrease of emissions during discharging.

The forecasting results of Fig. 9 and Fig. 10 present 13 curves, making the relevant charts illegible.  As the time-step in the two planning strategies subsequently presented is 1h, I suggest to provide forecasting results of 1h, 2h, 3h, 4h, 5h and 6h lags only so that only 7 curves appear on the same figure.  The 15min lag is nowhere used in the optimization that follows.

Author Response

Dear Reviewer,

Our manuscript, entitled “Bi-Objective Dispatch of Multi-Energy Virtual Power Plant: Deep-Learning based Prediction and Particle Swarm Optimization” (Manuscript ID: applsci-422324), was revised according to the reviewers’ comments. The itemized response to these comments based on our understanding is appended below and the revised manuscript as the Article is submitted. Many thanks for these valuable questions and the helpful suggestions. We gratefully appreciate your attention to our work and your guidance to our paper progressing.

We wish you a successful happy new year!

Reviewer 2 Report

The results section analyzes only wind power generation. However, according to Table 1, the system under analysis also contains PV generation. Please clarify why only wind has been considered.

It seems that Fig. 2 has been obtained somewhere else. If that is the case, proper reference must be included. Moreover, I would suggest redrawing this figure, so that its quality matches the other figures throughout the paper.

Lines 259-260: “[…] evolve from (23), but […]”

Do not split tables into 2 page

Line 353: “mono-objective”

Contractions should be avoided; for example, “it is” instead of “it’s”.

Author Response

Dear Reviewer,

Our manuscript, entitled “Bi-Objective Dispatch of Multi-Energy Virtual Power Plant: Deep-Learning based Prediction and Particle Swarm Optimization” (Manuscript ID: applsci-422324), was revised according to the reviewers’ comments. The itemized response to these comments based on our understanding is appended below and the revised manuscript as the Article is submitted. Many thanks for these valuable questions and the helpful suggestions. We gratefully appreciate your attention to our work and your guidance to our paper progressing.

We wish you a successful happy new year!

Round  2

Reviewer 1 Report

The authors have addressed all my comments.   I am happy with the current version of the manuscript.